# Exploring the Real-Time Variability and Complexity of Sitting Patterns in Office Workers with Non-Specific Chronic Spinal Pain and Pain-Free Individuals

**DOI:** 10.3390/s24144750

**Published:** 2024-07-22

**Authors:** Eduarda Oliosi, Afonso Júlio, Phillip Probst, Luís Silva, João Paulo Vilas-Boas, Ana Rita Pinheiro, Hugo Gamboa

**Affiliations:** 1Laboratory for Instrumentation, Biomedical Engineering and Radiation Physics (LIBPhys-UNL), Faculty of Sciences and Technology, NOVA University of Lisbon, 2820-001 Caparica, Portugal; a.julio@campus.fct.unl.pt (A.J.); p.probst@campus.fct.unl.pt (P.P.); lmd.silva@fct.unl.pt (L.S.); hgamboa@fct.unl.pt (H.G.); 2Research Centre in Physical Activity, Health and Leisure (CIAFEL-FADEUP), Faculty of Sport, University of Porto, 4000-000 Porto, Portugal; 3Centre for Research, Training, Innovation and Intervention in Sport (CIFI2D), Porto Biomechanics Laboratory (LABIOMEP), Faculty of Sport, University of Porto, 4000-000 Porto, Portugal; jpvb@fade.up.pt; 4Institute of Biomedicine (iBiMED), School of Health Sciences, University of Aveiro, 3800-000 Aveiro, Portugal; anaritapinheiro@ua.pt

**Keywords:** movement variability, motor adaptions, chronic pain, spinal pain, biomechanics, signal processing, nonlinear analysis

## Abstract

Chronic spinal pain (CSP) is a prevalent condition, and prolonged sitting at work can contribute to it. Ergonomic factors like this can cause changes in motor variability. Variability analysis is a useful method to measure changes in motor performance over time. When performing the same task multiple times, different performance patterns can be observed. This variability is intrinsic to all biological systems and is noticeable in human movement. This study aims to examine whether changes in movement variability and complexity during real-time office work are influenced by CSP. The hypothesis is that individuals with and without pain will have different responses to office work tasks. Six office workers without pain and ten with CSP participated in this study. Participant’s trunk movements were recorded during work for an entire week. Linear and nonlinear measures of trunk kinematic displacement were used to assess movement variability and complexity. A mixed ANOVA was utilized to compare changes in movement variability and complexity between the two groups. The effects indicate that pain-free participants showed more complex and less predictable trunk movements with a lower degree of structure and variability when compared to the participants suffering from CSP. The differences were particularly noticeable in fine movements.

## 1. Introduction

Chronic spinal pain (CSP), which includes low back pain (LBP), neck pain, and thoracic spinal pain, is a widespread global health issue [1], impacting millions of individuals [2]. CSP is a leading cause of disability, with LBP and neck pain being the most prevalent forms [3,4,5,6]. Recent research indicates that an estimated 619 million people experienced LBP in 2020, with projections suggesting an increase to 843 million by 2050 [7]. Job roles involving prolonged sitting posture, such as those in tax authorities, have been found to increase the risk of developing CSP due to ergonomic factors [7,8,9]. Tax authorities are engaged in tasks that necessitate extended periods of sitting posture, leading to heightened discomfort and resulting in changes in motor variability [8,10,11]. In addition, tax authority workers experience high levels of stress, which can affect their work performance, leading to reduced job satisfaction, health-related lost productivity, and increased absenteeism. These biopsychosocial risks can contribute to the onset or aggravation of ongoing CSP [12]. Hence, there is a growing concern regarding investigating occupational diseases in these workers.

The concept of a singular, correct posture has been challenged, making it imperative to adopt varying positions [13,14,15]. Furthermore, the relevance of variation, the study of variability, has become significant as it sheds light on how the motor system adapts to maintaining posture [16,17,18]. One of the most effective ways to measure changes in motor performance over time is through variability analysis. This method has gained attention recently due to its ability to identify variations in functional capacity and performance, ranging from healthy states to disability patterns [19]. In the field of kinesiology, the complexity of human movement is influenced by the interactions between various physiological systems. Understanding these interactions is crucial for maximizing an individual’s ability to function and perform, effectively responding to their environment [19,20,21]. Human motor variability can be described as the normal variations that occur in motor performance across multiple repetitions of the same tasks [22]. When executing the same task over multiple attempts, different performance patterns, including kinematic, kinetic, and muscle activation patterns, can be observed [23]. This variability is intrinsic to all biological systems and is readily observable. When a person tries to repeat the same movement twice, the two actions will never be identical or independent [19].

Previous research has revealed that prolonged sitting increases the variability in the displacement of the center of pressure (COP) and lumbar curvature in the anterior–posterior direction, while decreasing complexity represented by sample entropy, leading to discomfort [11]. Studies on trunk postural control using an unstable surface indicate that individuals with LBP have inferior control compared to those without LBP, as shown by higher root-mean-square displacement [24]. This impaired postural control may result from delayed or inaccurate corrective responses, increasing seat movements and potential trunk stiffening. A systematic review found that individuals with chronic LBP had delayed muscle onsets in response to perturbations, with inconclusive evidence on COP or kinematic differences compared to healthy controls [25]. Additionally, patients with non-specific neck pain or whiplash-associated disorder exhibit greater postural instability, linked to proprioceptive impairment rather than pain duration [26]. Older adults with neck pain show diminished functional performance, as indicated by wavelet analysis (but not for entropy) measures, compared to healthy controls [27]. It is interesting to note that both experimental and chronic neck–shoulder pain can increase variability in task timing, kinematics, and muscle activation during repetitive arm movement [28]. This suggests that variability plays an important role in motor control and may contribute to the transition from acute to chronic pain. During the acute stage, the central nervous system seeks the least painful biomechanical solution, while in the chronic stage, the chosen solutions are characterized by reduced flexibility of the motor system [28].

Therefore, it is crucial to understand the motor control of workers who sit for extended periods of time. One potential solution is to use inertial sensors that are embedded in smartphones to monitor these workers [29,30,31,32,33]. The present work is part of the Prevention of Occupational Disorders in Public Administration based on the Artificial Intelligence (PrevOccupAI) project. It involves stakeholders from biomedical engineering, health sciences, and public administration (tax authority) in Portugal. The main goal is to promote occupational health by identifying occupational risks. The focus is on characterizing and assessing daily working activities, working conditions, and potential risk factors associated with office work in the assistive technology field. The evaluation encompasses both the individual and organizational levels, utilizing non-intrusive sensing technology and questionnaires to accurately assess workers’ potential occupational risks within the tax authority. As part of the PrevOccupAI project, we are interested in studying motor control adaptation in individuals working for tax authorities and investigating CSP presence and perception. To achieve this, we studied the effects on trunk motor variability when working for five consecutive days, following the example of previous investigations that demonstrated changes in motor variability over time [11,28,34,35,36,37].

This study aimed to examine the real-time trunk dynamics in individuals with and without CSP while performing sedentary work tasks. The study had two main objectives: (a) to assess individual trunk motor variability of office workers in real time and (b) to test the hypothesis that individuals with and without pain have different responses to office work tasks.

## 2. Materials and Methods

### 2.1. Design

This cross-sectional study was performed at four locations of the Portuguese Tax and Customs Authority (Autoridade Tributaria e Aduaneira, AT) throughout the Lisbon Metropolitan Area. The study received ethical approval from the Universidade Nova de Lisboa Ethics Committee (No. CE/FCT/005/2022). All participants provided written informed consent.

### 2.2. Participants

The participants were selected at random as part of a campaign conducted by AT’s HR team. The following inclusion criteria were applied: adults 18 years of age and older without any associated pathology (e.g., neurological, orthopedic, rheumatic, oncological, or cardiorespiratory) and pregnancy.

The pain-free group (PG) (n = 6) was composed of individuals who had no history of back or lower limb pain or injury that restricted their ability to function or required medical attention. On the other hand, the chronic spinal pain group (CSPG) consisted of individuals who experienced non-specific spinal pain that persisted or reoccurred for a minimum of three months without any identifiable underlying cause (n = 10). This definition aligns with the criteria established by the International Association for the Study of Pain (IASP) and the ICD 11 guidelines [38,39].

Demographic information was collected through a single questionnaire to analyze the sample. The collected data included age (in years), sex (male or female), height (in centimeters), and mass (in kilograms), which was then used to calculate the Body Mass Index (BMI) using the formula: $BMI=\frac{mass\;(kg)}{{\left(height\;\left(m\right)\right)}^{2}}$. The International Physical Activity Questionnaire Short form (IPAQ-SF) [40] was included to evaluate physical activity. This survey consists of seven questions about the number of days and the amount of time spent doing vigorous and/or moderate physical activity (PA), walking, and sitting during weekdays. The final score is reported as low, medium, or high, and the level of physical activity per week through the metabolic equivalents [40]. The participants’ characteristics are summarized in Table 1. The data are expressed in terms of mean, standard deviation, or percentage.

### 2.3. Procedures

A series of data acquisitions were carried out over five consecutive days. Our previous published work [41] provides a detailed description of the acquisition protocol.

A cross-platform application developed specifically for the project was used for the data collection. The application creates unique user access for each participant, enabling them to complete questionnaires using either a computer or a smartphone. The application interface allows for scheduling acquisitions in advance, specifying the devices from which data should be acquired, as well as the start time and duration of the acquisition. Prior to the week of acquisitions, a preliminary session was held to inform the participants about the project, its objectives, and the acquisitions. Once informed consent was obtained, a profile was created for each participant. They were then given questionnaires, and the acquisition times were scheduled using the PrevOccupAI application, starting at the beginning and ending at the end of the workday. Participants were advised to schedule acquisitions while seated at their desks, preferably two in the morning and two in the afternoon.

At the beginning of each workday, the researchers attached the equipment to the workers, and the workers went about their daily tasks in their respective offices until the end of the day. After the workday was over, the equipment was removed and disinfected. The manuscript presented here contains part of a comprehensive analysis that collected multimodal information from multimodal sources.

During data collection, three different devices were used: two surface electromyography (EMG) sensors, a smartwatch, and a smartphone. The two EMG sensors (1000 Hz) were placed in the upper trapezius, while the smartwatch, which recorded the heart rate (1 Hz), was placed on the right wrist. The smartphone was positioned on the subject’s chest using a harness (Figure 1).

It is important to note that the placement of the EMG sensors and smartwatches did not affect the positioning of the smartphone sensors. Inertial sensors from smartphones were chosen as the most suitable equipment to capture trunk posture data from subjects. The use of smartphones was deemed appropriate due to their widespread availability and the fact that they are equipped with a range of sensors.

Thus, the accelerometer (ACC), magnetometer (MAG), gyroscope (GYR), and rotation vector (RV) data were acquired using a Xiaomi Redmi Note 9. The Redmi Note 9 runs the Android operating system. The Android OS restricts the ACC, GYR, and RV to 100 Hz, while the MAG is sampled at 50 Hz.

### 2.4. Data Analysis

#### 2.4.1. Data Synchronization

Android’s battery-saving mechanisms can cause sensor data to be sampled at different times and with irregular intervals. Due to this reason, the following two-step procedure was performed:

*Determining start and stop points:* The starting point was defined as the initial timestamp of the last sensor to begin acquisition, and the stopping point as the final timestamp of the first sensor to cease acquisition. All signals were cropped to this range.

*Resampling:* Each signal was individually resampled to 100 Hz to ensure a consistent and uniform sampling frequency. This was achieved by creating a new time axis with constant intervals. Each component of each signal was then linearly interpolated according to this new time axis.

#### 2.4.2. Accelerometer and Rotation Vector Pre-Processing

The following pipeline was used to process accelerometer data:

1. High frequencies’ removal through a lowpass filter with 10 Hz.

2. Gravity component removal.

3. Detrending by subtracting the mean of the signal from each sample.

4. Moving average smoothing filter with a 150-sample window.

Regarding the rotation vector, a moving average smoothing filter with a window of 5 samples was employed to eliminate noise arising from factors like sensor inaccuracies. This filter reduced the impact of noise by averaging out fluctuations and high-frequency variations in the signal.

5. The signals were segmented due to the large length of the time series (approximately 5 h).

To achieve this segmentation, a 15-min window size was chosen. The central window was labeled as “Lunch” time, with the windows immediately before and after also classified as “Lunch” breaks. The windows before “Lunch” were tagged as “AMx”, where ‘x’ denotes the window number, and the same labeling scheme was applied to the windows following “Lunch”, which were labeled as “PMx” (Table 2).

An algorithm was used to detect and remove periods when workers were not at their desks or transitioning between postures that were outside the analysis range [42]. This algorithm was fed by ACC data and considered a threshold to identify postural changes and walking intervals. The threshold was determined based on the magnitude of acceleration (mag) from the X, Y, and Z axes. The magnitude (mag) represents the combined acceleration value along the x, y, and z axes. In Figure 2, the graph shows the signal with green highlights indicating the segments that were removed.
(1)$$\mathrm{mag}=\sqrt{x_{\mathrm{acc}}^{2}+y_{\mathrm{acc}}^{2}+z_{\mathrm{acc}}^{2}}$$

The threshold was based on a human walking accelerometry analysis previously documented in [43]. A walking velocity of 0.55 m/s was considered as representative of a working scenario. Using acceleration values in the x, y, and z coordinates from an ACC placed on the C7 vertebrae ($a_{x}$ = 1.02 ${m/s}^{2}$; $a_{y}$ = 1.15 ${m/s}^{2}$; $a_{z}$ = 1.41 m/s^2^), the following equation calculated the acceleration magnitude threshold:(2)$$\mathrm{mag}=\sqrt{1.02^{2}+1.15^{2}+1.41^{2}}\Leftrightarrow \mathrm{mag}=2.08\;\left({m/s}^{2}\right)$$

The result presented an acceleration magnitude threshold of approximately 2 m/s^2^. This value was also corroborated by the calculation of the same threshold when establishing the relationship between energy consumption (metabolic equivalent, MET) and acceleration magnitude [44,45] with a MET for walking, gathering things (encompassing the movement that can be considered as postural transitions), and 3 METs for getting ready to leave in occupational situations. The threshold was determined by:(3)$$\mathrm{MET}=5.289*\mathrm{mag}-8.5548\Leftrightarrow \mathrm{mag}=\frac{3+8.5548}{5.289}\Leftrightarrow \mathrm{mag}=2\;\left({m/s}^{2}\right)$$

Both approaches yielded a similar approximate value for the acceleration magnitude threshold of 2 m/s^2^, and this was the threshold defined for non-seated interval detection. Consequently, the algorithm removed signal intervals exceeding that threshold, preserving only the signal portions containing the sway variability in the postural data for analysis.

Quaternions were utilized to estimate the COP projection in the xy plane, which was derived from the RV. The algorithm detected and recorded the indexes of intervals that needed to be removed. The removal of these intervals by their indexes was then applied to the rotation vector signal, synchronizing the data and cleaning the signal of non-seated periods and transitions between postures.

Subsequently, the quaternions obtained from the RV were transformed into Euler angles. The median of each Euler angle was subtracted to establish a reference position centered at the origin (0,0). These angles were then converted into positions in the xz-plane, with the phone placed on the chest near the sternum (Figure 3). This process enabled the estimation of the COP projection in the horizontal plane that represented the subject’s movement during work. The COP projection comprised two time series, namely COPx (anterior–posterior) and COPy (medial–lateral) (Figure 4).

To ensure that all participants were in the same postural sway condition, a limited zone of movement was defined according to the literature [42]. An ellipse was created to enclose the postural sway signal, with dimensions of AP radius = 25 mm and ML radius = 18 mm. The signal inside this ellipse was the one extracted for further analysis. After excluding walking periods, the rotation vector time series was used to transform the coordinates into Euler angles. These angles were then converted into projection coordinates to obtain the displacement value in millimeters.

### 2.5. Measures of Variability

#### 2.5.1. Linear Measures

After generating the final dataset containing the COP time series, we extracted a set of linear measures to evaluate the movement patterns of the trunk and the signal characteristics for each participant. To calculate these linear parameters, the TSFEL Python package, which is a library described in [46], was utilized. The TSFEL package helped us to compute various metrics that were considered in this analysis. Table 3 provides a comprehensive list of these metrics, along with their interpretations when applied to COP time series data. Additionally, specific parameters related to postural sway were calculated. Table 4 presents the details of these parameters, including their units and the formulas used for their calculation.

Linear measures of variability are crucial for providing information about the amount of movement and variability around a mean value. However, it is important to note that variability is an inherent characteristic of all biological systems and describes the normal variations that occur in motor performance across multiple repetitions of a task. While traditional linear tools such as standard deviation (SD), coefficient of variation (CV), and range may provide valuable insights, they assume that each cycle of movement is independent of past and future cycles, and that variations between cycles are random. This assumption may not be accurate, and traditional tools may provide different results when compared with nonlinear measures [16,47].

**Table 3 sensors-24-04750-t003:** Linear parameters of the time series calculated using TSFEL. Adapted from [46].

Parameter	Description	Units
**Statistical Domain**		
Maximum	Computes the maximum value of the signal.	m/s^2^
Minimum	Computes the minimum value of the signal.	m/s^2^
Mean	Computes the mean value of the signal.	m/s^2^
Standard deviation	Computes the SD of the signal.	m/s^2^
Root mean square	Computes the RMS of the signal.	m/s^2^
Interquartile range	Computes the IQR of the signal.	m/s^2^
Variance	Computes the variance of the signal.	(m/s^2^)^2^
**Temporal Domain**		
Autocorrelation	Computes the autocorrelation of the signal.	-
Signal distance	Computes the time series traveled distance.	-

**Table 4 sensors-24-04750-t004:** Linear measures to assess the sway variability in trunk movement patterns [48].

Metric	Formula	Units
Sway area	$\sum_{n}\left|X_{n+1}\cdot Y_{n}-X_{n}\cdot Y_{n+1}\right|$	mm^2^
Sway range	$\sqrt{{\left(max\left(X\right)-min\left(X\right)\right)}^{2}+{\left(max\left(Y\right)-min\left(Y\right)\right)}^{2}}$	mm
Sway range (AP)	$\left|max(X)-min(X)\right|$	mm
Sway range (ML)	$\left|max(Y)-min(Y)\right|$	mm
Sway path	$\sum_{n}\sqrt{{\left(X_{n+1}-X_{n}\right)}^{2}+{\left(Y_{n+1}-Y_{n}\right)}^{2}}$	mm
Sway distance (RMS)	$\sqrt{\frac{1}{N}\sum_{n}\left(X^{2}+Y^{2}\right)}$	mm
Mean sway velocity	$\frac{fs}{N}\sum_{n}\sqrt{{\left(X_{n+1}-X_{n}\right)}^{2}+{\left(Y_{n+1}-Y_{n}\right)}^{2}}$	mm/s

#### 2.5.2. Nonlinear Measures

To gain a more complete understanding of how a movement pattern changes over time, it is critical to also include nonlinear techniques in the analysis. Traditional linear measures can miss certain valuable insights that nonlinear analysis can provide, which can help us better understand the actual structure of variability [16,47]. When it comes to analyzing trunk movement patterns, it is important to assess complexity using nonlinear parameters. This helps to reveal the variability in the pattern over time by examining the entire time series [20]. In this study, two nonlinear measures were employed to evaluate the complexity of variability:

*Multifractal Detrended Fluctuation Analysis (MDFA):* Fractal measures are an effective way to measure the complexity of fluctuations in a system. Multifractal Detrended Fluctuation Analysis (MFDFA) is a method that takes into account both small and large oscillations in a time series, which contain important information. The Hurst exponent (*H*) is a measure of the correlation among data points in a time series. The DFA method includes a parameter called “*q*” that allows for the analysis of different statistical moments of the fluctuation function. Each *H* value for a specific “q” order provides insights into how the scaling behavior of the time series changes with different statistical properties of the fluctuations. When considering MFDFA, “*Hq*” represents the Hurst exponent calculated for different orders “*q*”. These values are dimensionless and usually range between 0 and 1. They are scale-invariant measures that describe the fractal nature of the data and do not have units like meters or seconds [49,50,51].

A q-order range spanning from −5 to 5 was employed for the analysis of the time series, utilizing a polynomial order of 0.5. To enhance the robustness of our analysis, a set of scaling windows was implemented, ranging from 5 to the length of the time series divided by 8, with linearly spaced steps, resulting in a total of 30 windows.

The calculation of the *H* and multifractal spectrogram was executed using the Fathon Python library [52]. Notably, the countbysteps parameter in our code determined the number of windows used for computing *H* values. The determination of the last windows to be included was based on the total length of the time series divided by 8, resulting in 30 windows. The computation of Hurst exponent values involved varying window sizes. Specifically, we initiated the process with 5 windows and incrementally increased the count by 5 steps until reaching a total of 30 windows (e.g., 5, 10, 15, 20, 25, 30 windows). This stepwise approach allowed us to comprehensively assess the multifractal characteristics of the time series across different window configurations. The metrics are compiled in Table 5.

To ensure that the time series captured the phenomenon of interest and was not derived from a stochastic process, we performed a surrogate analysis. For each time series (3 × 2 (morning and afternoon) × 3 days), we generated 100 surrogates using the Iterated Amplitude-Adjusted Fourier Transform (IAAFT) algorithm [53]. H(q)=0 was used since it is the order used in the traditional DFA. We then verified if the value of H(q)=0 for the original time series was inside or outside of the 95% interval of confidence. If the value was outside, we considered genuine fractal characteristics were present in the original time series.

*Sample Entropy (SampEn):* SampEn is a nonlinear measure to quantify the regularity or unpredictability of time series data. It quantifies the likelihood of finding repeated patterns of a certain length in the data. A higher SampEn value indicates greater irregularity or complexity, suggesting that the time series is less predictable or more disordered. Conversely, a lower SampEn value suggests more regularity or predictability in the data. Similar to the Hurst Exponent, SampEn was computed for the original time series and surrogates, using the EntropyHub Python library [54].

The SampEn was calculated with an embedded dimension of 4, as per the recommendation in the literature [55]. It was observed that nearly 98% of the original data points were outside the limit interval. This reaffirmed the fact that the time series exhibited authentic fractal behavior that was absent in the surrogate data.

### 2.6. Statistical Analysis

Descriptive statistics were used to summarize the numerical variables, specifically the mean and standard deviation. The data displayed a normal distribution, which was confirmed through the Shapiro–Wilk test. A mixed ANOVA (Analysis of Variance) was utilized to compare the average group values of all the calculated measures in order to analyze the impact of the days and periods as independent variables on all the potential dependent variables. Whenever there was a significant difference within or between groups (*p* < 0.05), the effect size of the ANOVA was computed using partial eta squared (${n_{p}}^{2}$) and categorized as small (<0.06), medium (0.06–0.14), and large (>0.14) [56,57].

## 3. Results

### 3.1. Measures of Magnitude of Variability

Table 6 presents the results of the time series analysis for linear measures, along with the corresponding factors. The table also features the *p* values resulting from the comparison tests carried out both between and within subjects. Table 7 showcases the results of the ANOVA for linear measures, emphasizing the statistically significant differences, effect sizes, and their interactions.

Results from the statistical analysis conducted on the linear measures showed some interesting findings, indicating significant differences across various metrics, particularly for those with larger effect sizes. Maximum ML (m/s^2^) and signal distance AP and ML exhibited significant differences within subjects. Between groups, several metrics showed significant variations, including the minimum value of the signal AP (mm^2^), SD AP (m/s^2^), variance AP ((m/s^2^)^2^), RMS ML (m/s^2^), and IQR AP (m/s^2^). It is noteworthy that most of these values were higher in the PG. The summary of findings is presented in Table 7.

Throughout the week, there were significant differences in the “maximum (ML direction)” values of the PG, but not in the CSPG. Specifically, the maximum ML metric displayed a large effect size ($n_{p}^{2}$ = 0.149) with *p* = 0.041 in the PG. On the other hand, the metric minimum AP mm^2^ differed notably between the groups, with more negative values in the CSPG, indicating the patterns of most were on the posterior side. The effect size was large ($n_{p}^{2}$ = 0.386) with *p* = 0.010. There were also significant between-group differences in SD (AP), with the CSPG values consistently higher, indicating greater displacement toward the front side. The effect size was large ($n_{p}^{2}$ = 0.534) with *p* = 0.001.

The variance in the AP direction showed a contrasting trend between the PG and the CSPG. The PG tended to decrease over the week, while the CSPG showed an increase. This variation yielded a statistically significant effect size of group ($n_{p}^{2}$ = 0.505) with *p* = 0.002. The RMS in the ML direction metric also showed a statistically significant effect size of group ($n_{p}^{2}$ = 0.388) with *p* = 0.010. Moreover, the IQR in the AP direction demonstrated significant differences between groups. In general, there was a decrease in both groups as the week progressed. The effect size was large ($n_{p}^{2}$ = 0.398) with *p* = 0.009.

There were no significant differences between the PG and CSPG values in both AP and ML components of the signal distance. However, a significant effect was found between the days of the week within groups, with both groups demonstrating an increase in the values as the day progressed. Precisely, signal distance AP (week) had a large effect size ($n_{p}^{2}$ = 0.380) with *p* = 0.001 and signal distance ML (week) had a large effect size ($n_{p}^{2}$ = 0.383) with *p* = 0.001. Lastly, there were no significant differences (*p* > 0.05) in the linear metrics of postural sway between the groups in any direction.

### 3.2. Measures of Structure of Variability

Table 8 presents the descriptive statistics of 18 outcomes of the time series analysis for nonlinear measures, along with the corresponding factors of days and periods. The table also displays the *p* values resulting from comparison tests conducted within and between subjects. The ANOVA results for nonlinear measures are shown in Table 7, highlighting the statistically significant differences, effect sizes, and their interactions.

H(q)=−5 in the AP direction showed significant differences between groups (${n_{p}}^{2}=0.281$, p=0.035, large effect size), indicating noteworthy divergence. Similarly, H(q)=0 in the AP direction showed significant differences between groups (${n_{p}}^{2}=0.257$, p=0.045, large effect size), highlighting distinct patterns in postural dynamics.

Despite a smaller effect size, H(q)=4.5 in the AP direction revealed significant differences along the days of the week within the CSPG compared to the PG (${n_{p}}^{2}$ = 0.0319, *p* = 0.023), suggesting consequential variations over time in the CSPG.

Both the AP and ML components of negative width showed significant differences between groups (AP: ${n_{p}}^{2}=0.315$, p=0.024; ML: ${n_{p}}^{2}=0.324$, p=0.021, both large effect sizes) (Figure 5). Moreover, within the PG, both components exhibited an increase throughout the day, indicating temporal trends in COP dynamics among individuals without spinal pain. Interestingly, both groups displayed similar SampEn values in both directions, and these values did not show statistically significant differences (p>0.05).

## 4. Discussion

The main objective of this research was to explore the impact of office work adaptations on the trunk posture in people suffering from non-specific spinal pain in comparison with healthy individuals. The analysis of movement data in individuals with pain (CSPG) versus those without pain (PG) during sitting and working revealed significant differences in both statistical and temporal domain metrics, highlighting variations in movement patterns and intensity between the two groups.

### 4.1. Statistical Domain Metrics

Statistical domain metrics showed distinct differences in trunk movement data distribution and spread. Maximum acceleration values indicated peak movement intensity, with significant differences in the ML direction. Minimum acceleration values were more negative in the CSPG, particularly in the AP direction, suggesting a greater tendency for right-side trunk fluctuations in individuals with pain. The SD of trunk movements, reflecting variability, was higher in the CSPG in the AP direction, indicating more dispersed movements. RMS values, measuring overall movement magnitude, were higher in the CSPG in the ML direction, suggesting more energetic movements. The IQR, representing the middle 50% of the data, was also higher in the CSPG in the AP direction, supporting increased variability in trunk movements. The variance of the signal indicated more consistent and varied movements in the CSPG in the AP direction. The linear metric, SD AP m/s^2^, had the most significant effect size between groups (${n_{p}}^{2}$ = 0.534), demonstrating considerable differences between the PG and CSPG across multiple variables related to postural dynamics.

### 4.2. Temporal Domain Metrics

Movement patterns and cumulative aspects over time in the temporal domain metrics were analyzed. The signal’s autocorrelation, which measures its correlation with a lagged version of itself, showed no significant differences between groups. This suggests repetitive sitting posture movements were similar in individuals with and without pain. However, the signal distance, reflecting the total trunk movement over time, was significantly different in both directions within subjects. This indicates distinct cumulative trunk displacement between the PG and CSPG, suggesting variations in overall movement patterns during sitting and working.

### 4.3. Signal Structure and Complexity

The structure of the signal signature varied between groups but remained consistent throughout the week. For H(q)=−5, emphasizing the smallest fluctuations [49], *H* was different between groups in the AP direction, with higher values in the CSPG, indicating greater variability in fine trunk movements. For q=0, *H* was different between groups in the AP direction, indicating higher overall variability in the CSPG. For q=4.5, focusing on the largest fluctuations [49], H(q) was different between groups in the AP direction, highlighting more significant deviations and large-scale movements in the CSPG. In monofractal analysis (H(q)=2), results were equal between groups in both AP and ML directions, suggesting consistent overall trunk movement complexity across scales for both groups. The negative width, indicating fine-scale movement variability, was higher in the CSPG in both directions, supporting more intricate control mechanisms for minor corrections in individuals with pain.

Postural variables derived from COP provide insights into system complexity through time series analysis. The time intervals displayed a non-random pattern, adhering to an inverse power law distribution (1/fα), indicating the presence of pink noise or long-range correlations. Past studies using DFA analysis affirmed the existence of physiological complexity [50,58]. Results showed that PG values were closer to 1.5 (indicating Brown noise) compared to CSP values, which were closer to 2.0, suggesting better control over body movements in healthy individuals. CSP individuals showed a tendency to continue moving in the same direction, indicating a persistent trend [59]. The study also found that *H* seemed to distinguish between healthy subjects and those with CSP, similar to previous studies with other chronic pain conditions [51]. Additionally, the results support previous findings that individuals with CSP tend to have a more static sitting behavior [60].

The DFA method has recently gained attention for assessing fractal scaling characteristics and identifying long-range correlations in noisy, nonstationary time series. Multifractal analysis characterizes scaling at various statistical moments *q*. Negative values of *q* amplify small fluctuations, while positive moments amplify large fluctuations [61]. This method has been useful in various fields, including human gait [62,63], simulated driving tasks [64], pain dynamics [65], neurological physical therapy [21], and concurrent motor tasks [66]. One reason to use DFA is to avoid falsely detecting correlations caused by nonstationarities in the time series [49]. Previous studies have revealed an array of findings regarding postural sway. For instance, quiet standing in healthy young adults exhibits fractal fluctuations over time scales from 10 s to 10 min in both ML and AP directions, with alpha exponents ranging from 0.68 to 1.47 [59,67].

Delignières et al. found that the short-term slope in quiet stance was AP = 1.65 and ML = 1.70, and the long-term slope was AP = 1.22 and ML = 1.00 in male volunteers [68]. The research used a multifractal formalism to study movement control adaptation to dual motor tasks, examining the global (monofractal) scaling related to long-range correlations and long-term memory of the signal in H(q)=2. The degree of multifractality was calculated as H(q) = −15 and H(q)=15 to assess scaling differences for small and large fluctuations. When grouped together, the H(q)=2 scaling exponents were 1.02 ± 0.09 for cycling, 0.59 ± 0.09 for finger tapping, and 0.76 ± 0.11 for circling. The study discovered a significant difference across these motor-task-specific exponents [66].

In a study conducted by Lau et al. (2012), it was found that patients with neck pain exhibited smaller short-term Hurst exponent values than healthy subjects across all four phases: “self-adopted posture” (neck pain: H = 0.73 vs. healthy: 0.78), “upright posture before training” (neck pain: H = 0.81 vs. healthy: 0.85), “training stage” (neck pain: H = 0.76 vs. healthy: 0.83), and “upright posture after training” (neck pain: H = 0.82 vs. healthy: 0.86) [69]. Additionally, data processing techniques appeared to have no effect on the magnitude of variability (SD and RMS), but did affect the structure of variability (DFA α and SampEn) in COP displacement, as discussed in previous research [70].

Our previous research on office workers indicated significant differences in postural changes based on time of day and day of the week, with fatigue leading to increased movement and postural variability [71]. This study’s novelty lies in comparing the CSPG and healthy controls over a week, using smartphone sensors adaptable to other contexts. Smartphones were chosen for their ability to interface with the PrevOccupAI application and collect data from multiple workers simultaneously, creating realistic working conditions.

### 4.4. Implications and Future Research

Contrary findings in CSP patients during seated tasks suggest complex postural adaptations [72]. CSP’s etiology is multifactorial, with only a portion attributable to sedentary risks [73]. A recent study distinguished CSP patients from asymptomatic participants using an experimental protocol assessing 72 movement biomarkers [74]. Research employing a two-group design revealed differential effects of cognitive dual-tasks on postural sway in CSP patients versus controls [75]. Further, CSP impacts postural control, with affected individuals showing divergent trunk muscle activation patterns, influencing tissue loading and requiring further investigation [76,77]. Mahdavi et al.’s meta-analysis highlighted the heightened CSP risk associated with prolonged sitting and extended driving periods, emphasizing the need for comprehensive CSP research. The present study did not account for uncontrolled variables, such as commuting time, which could impact outcomes [73]. Future studies should incorporate these variables.

The association between sitting time and CSP has shown inconsistent findings due to variations in study design, measurement methods, and occupational group heterogeneity. This study only included public administration office workers, limiting the generalizability of results to other populations. This study used cross-sectional data, limiting the ability to determine causality between motor control changes, sitting, and CSP. Future research could explore the link between perceived threats and protective responses [78]. Understanding small posture changes is vital for designing workplace guidelines to prevent injury and validating models of spinal pain risk factors [79]. Lifestyle interventions, including technology use, motivational counseling, and self-monitoring, have reduced sedentary behavior among individuals with various medical conditions. Addressing sedentary behavior as a risk factor in health education is recommended to reduce CSP prevalence [73]. Modern ergonomics must evolve to manage complex human-system interactions [79]. Future research should analyze electromyographic parameters [25,80] and categorize patients into different severity risk groups [81]. Finally, future studies should consider comparing the COP estimation method. Although based on established physical principles and developed internally by our research team, future investigations should proceed with a validation analysis through comparative studies or external benchmarks.

## 5. Conclusions

This study found new evidence linking CSP to trunk posture, especially during prolonged sitting. Differences were notable in fine movements, indicated by higher negative width in the CSPG. The analysis of the generalized Hurst exponent highlighted the multifractal nature of postural sway in the AP direction, with the CSPG showing higher values across all orders. Further research is needed to determine if the statistically significant differences found are clinically meaningful.

## Figures and Tables

**Figure 1 sensors-24-04750-f001:**
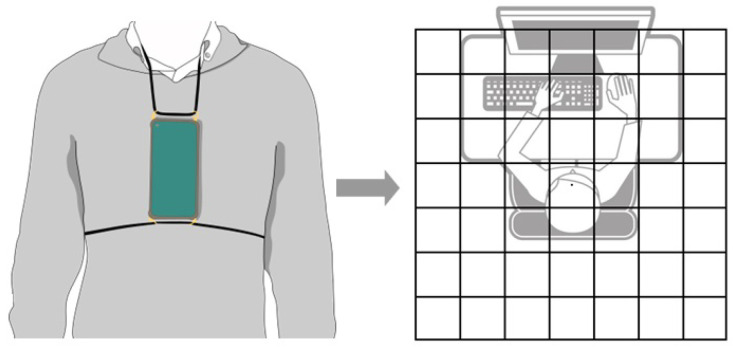
Postural analysis using a smartphone mounted on the chest.

**Figure 2 sensors-24-04750-f002:**
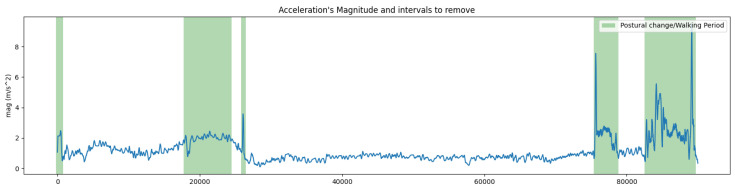
Detection of periods of interest.

**Figure 3 sensors-24-04750-f003:**
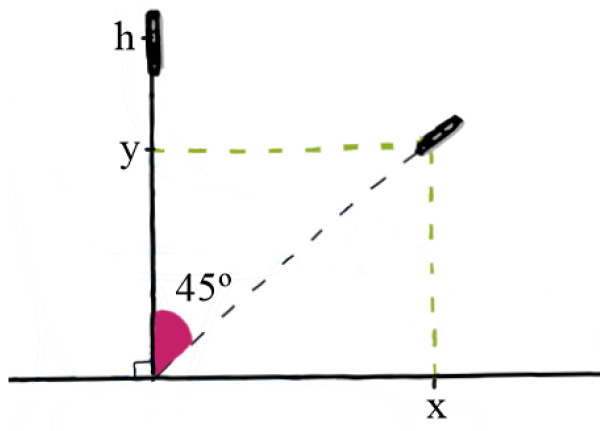
Decomposition-based approach for the projection’s calculation of displacement.

**Figure 4 sensors-24-04750-f004:**
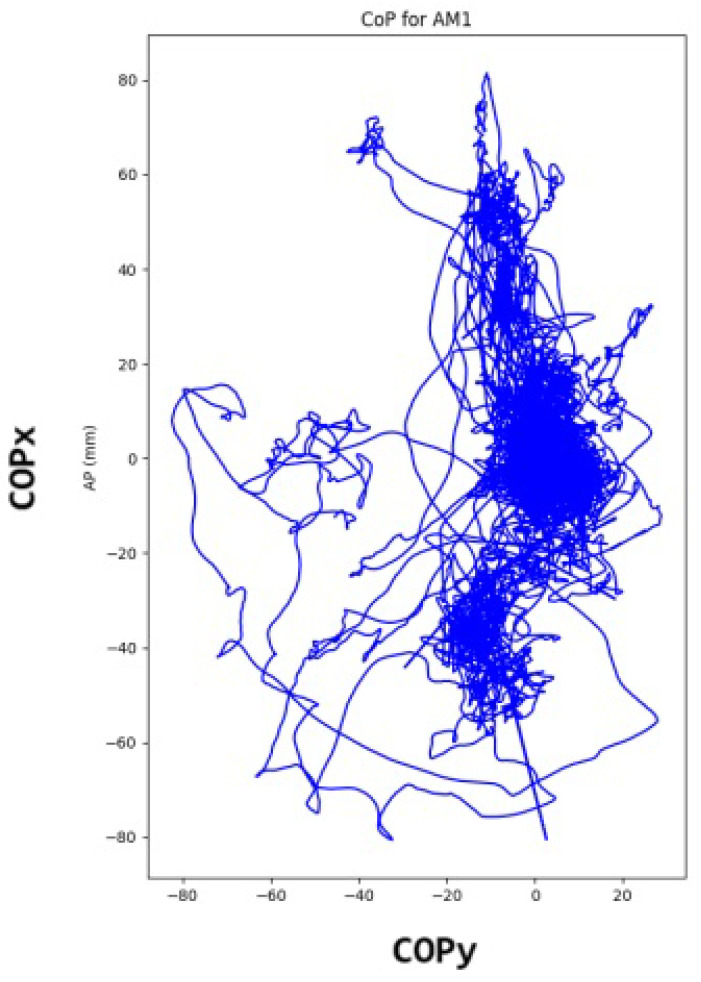
Data from two anatomical axes.

**Figure 5 sensors-24-04750-f005:**
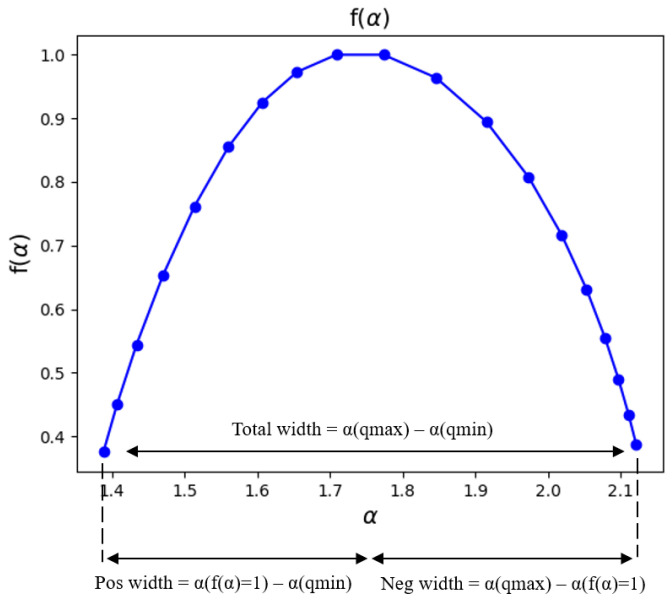
Multifractal spectrum. The arrow indicates the difference between the maximum and minimum hq that is called the multifractal spectrum width.

**Table 1 sensors-24-04750-t001:** Participants’ characteristics.

Variable	PG	CSPG
Sex (female, %)	50.00	80.00
Age	51.33 (4.41)	54.00 (6.51)
BMI (kg/m^2^)	28.86 (3.87)	26.75 (6.14)
Years of profession	21.17 (13.64)	18.16 (13.91)
Time working per week (hours)	35.42 (3.32)	41.10 (7.07)
Time sitting per day (week)	8.28 (2.13)	9.20 (3.39)
Time sitting per day (weekend)	3.61 (1.65)	4.18 (1.78)
**Physical activity** (%)		
Low	33.33	30.00
Moderate	50.00	40.00
High	16.70	30.00

**Table 2 sensors-24-04750-t002:** Time series lengths for each window.

Window	Length
AM1	1289
AM2	1167
AM3	1171
PM1	1589
PM2	1126
PM3	1246

**Table 5 sensors-24-04750-t005:** Nonlinear parameters calculated using Multifractal Detrended Fluctuation Analysis.

Parameter	Description
$H_{qs\;=\;-5}$	Computes H for q-order = −5.
$H_{qs\;=\;0}$	Computes H for q-order = 0.
$H_{qs\;=\;2}$	Computes H for q-order = 2.
$H_{qs\;=\;4.5}$	Computes H for q-order = 4.5.
Total width	Computes the width of multifractal spectrum.
Positive width	Computes the left width of the multifractal spectrum.
Negative width	Computes the right width of the multifractal spectrum.
Maximum spectrum	Computes maximum value of the multifractal spectrum.

**Table 6 sensors-24-04750-t006:** Mean ± SD values for the calculated linear measures and their comparisons.

Metric						Day/Time							WS	BS
	**PG**						**CSPG**						* **p** *	* **p** *
	**Day 1**		**Day 3**		**Day 5**		**Day 1**		**Day 3**		**Day 5**			
	**AM**	**PM**	**AM**	**PM**	**AM**	**PM**	**AM**	**PM**	**AM**	**PM**	**AM**	**PM**		
Sway area	53.98 ± 25.92	49.52 ± 7.05	38.06 ± 11.54	47.17 ± 6.18	51.20 ± 22.75	44.12 ± 15.94	49.83 ± 18.92	60.29 ± 28.24	45.92 ± 21.45	45.35 ± 14.11	53.04 ± 20.02	47.67 ± 23.14	0.37	0.61
Sway range	29.80 ± 6.26	30.69 ± 3.96	28.80 ± 3.95	31.14 ± 6.23	30.13 ± 10.07	29.39 ± 5.91	25.84 ± 6.55	30.17 ± 6.63	24.44 ± 8.04	29.00 ± 9.58	26.87 ± 8.38	24.60 ± 10.73	0.56	0.20
Sway range (AP)	22.79 ± 3.80	23.55 ± 4.58	22.50 ± 3.58	24.21 ± 3.82	24.41 ± 8.39	22.94 ± 7.23	20.73 ± 6.25	24.47 ± 6.68	19.08 ± 6.66	22.03 ± 8.28	21.84 ± 7.03	19.22 ± 8.70	0.61	0.32
Sway range (ML)	17.73 ± 6.01	18.38 ± 1.76	16.02 ± 3.24	18.67 ± 5.90	16.54 ± 8.03	16.73 ± 5.24	14.03 ± 4.94	16.87 ± 3.63	14.62 ± 5.49	17.40 ± 6.77	14.36 ± 5.79	14.30 ± 6.43	0.53	0.20
Sway path	29.80 ± 6.26	49.52 ± 7.05	28.80 ± 3.95	47.17 ± 6.18	30.13 ± 10.07	44.12 ± 15.94	25.84 ± 6.55	60.29 ± 28.24	24.44 ± 8.04	45.35 ± 14.11	26.87 ± 8.38	47.67 ± 23.14	0.44	0.47
Sway distance	10.09 ± 0.85	9.66 ± 1.38	9.92 ± 1.78	10.51 ± 2.66	10.433 ± 3.58	9.31 ± 2.80	9.88 ± 3.60	10.51 ± 1.42	8.31 ± 3.06	9.40 ± 2.34	9.66 ± 2.24	9.45 ± 4.00	0.87	0.53
Sway velocity	16.90 ± 7.83	15.22 ± 2.20	13.51 ± 3.45	15.45 ± 3.54	14.95 ± 6.16	15.04 ± 5.81	16.91 ± 7.61	19.90 ± 9.30	15.24 ± 6.42	16.02 ± 4.69	16.41 ± 6.67	15.08 ± 6.89	0.69	0.47
Maximum (AP)	12.39 ± 7.66	11.63 ± 2.80	9.26 ± 8.77	8.97 ± 7.59	10.04 ± 6.21	9.77 ± 7.44	8.58 ± 4.65	11.45 ± 4.56	9.97 ± 3.27	11.81 ± 7.24	14.89 ± 4.27	15.12 ± 2.20	0.62	0.24
Maximum (ML)	6.83 ± 3.81	10.90 ± 1.99	4.61 ± 4.15	8.47 ± 6.08	5.40 ± 6.37	3.34 ± 3.20	8.84 ± 5.05	8.92 ± 2.38	6.11 ± 3.62	8.59 ± 4.29	9.47 ± 3.93	9.75 ± 4.46	**0.04 ***	0.11
Minimum (AP)	−8.51 ± 3.81	−10.60 ± 5.89	−8.14 ± 5.05	−10.58 ± 4.58	−7.07 ± 9.11	−4.52 ± 7.07	−13.28 ± 5.32	−13.81 ± 4.84	−12.18 ± 3.76	−13.02 ± 7.14	−11.32 ± 3.87	−9.29 ± 5.98	0.11	**0.01 ***
Minimum (ML)	−8.27 ± 1.40	−7.13 ± 4.55	−8.26 ± 4.69	−9.77 ± 3.93	−6.63 ± 3.46	−6.79 ± 4.13	−6.76 ± 6.22	−8.16 ± 3.55	−10.39 ± 4.02	−9.07 ± 3.34	−7.59 ± 4.53	−8.51 ± 2.92	0.46	0.51
Mean (AP)	1.65 ± 4.37	0.76 ± 4.32	−0.30 ± 5.03	0.04 ± 5.25	1.97 ± 5.04	3.73 ± 5.34	−1.90 ± 4.02	−0.12 ± 4.28	0.44 ± 2.94	0.11 ± 5.64	2.11 ± 4.36	2.59 ± 3.37	0.21	0.48
Mean (ML)	−1.59 ± 1.56	2.11 ± 3.50	−2.33 ± 3.04	−0.69 ± 2.48	0.057 ± 3.92	−1.38 ± 2.47	1.55 ± 5.40	0.08 ± 2.82	−2.03 ± 2.50	−0.07 ± 2.48	0.08 ± 2.79	0.00 ± 3.33	0.14	0.44
Standard deviation (AP)	4.10 ± 1.73	3.78 ± 0.89	3.13 ± 2.01	3.52 ± 1.21	3.12 ± 1.33	2.32 ± 1.55	4.39 ± 1.14	5.09 ± 0.81	4.01 ± 0.95	4.75 ± 1.08	4.99 ± 1.58	4.72 ± 1.00	0.22	**0.001 ***
Standard deviation (ML)	2.58 ± 1.25	2.88 ± 0.64	1.95 ± 1.11	2.84 ± 1.77	2.07 ± 1.18	1.62 ± 1.06	2.68 ± 0.78	2.98 ± 0.39	2.83 ± 0.83	2.98 ± 0.96	3.07 ± 1.22	3.20 ± 1.00	0.38	0.07
Variance (AP)	22.21 ± 13.96	17.22 ± 6.44	14.65 ± 14.41	16.25 ± 9.11	15.75 ± 10.28	9.09 ± 8.98	24.52 ± 11.77	32.45 ± 8.86	19.41 ± 8.22	27.02 ± 11.40	31.13 ± 16.82	27.13 ± 10.41	0.28	**0.002 ***
Variance (ML)	9.12 ± 8.97	9.85 ± 3.62	5.46 ± 4.45	13.02 ± 14.51	6.40 ± 4.45	5.07 ± 6.29	10.23 ± 5.04	9.97 ± 2.71	10.48 ± 5.58	10.71 ± 6.20	12.60 ± 9.12	13.14 ± 8.41	0.69	0.25
Root mean square (AP)	6.77 ± 3.04	6.68 ± 1.56	5.87 ± 3.63	6.75 ± 2.00	6.82 ± 2.99	7.39 ± 5.14	7.11 ± 2.59	8.25 ± 1.95	7.72 ± 2.34	8.73 ± 2.12	8.56 ± 2.13	7.71 ± 2.21	0.87	0.06
Root mean square (ML)	5.16 ± 2.66	6.20 ± 1.54	4.06 ± 2.27	4.72 ± 1.83	4.94 ± 1.86	3.80 ± 2.25	6.94 ± 2.09	5.66 ± 1.15	4.98 ± 1.65	5.16 ± 0.88	5.77 ± 1.74	5.50 ± 1.68	0.12	**0.01 ***
Interquartile range (AP)	5.12 ± 2.97	4.11 ± 1.71	3.61 ± 2.99	4.33 ± 1.31	3.83 ± 1.57	2.67 ± 2.34	6.13 ± 2.40	7.14 ± 1.33	5.01 ± 1.93	6.49 ± 1.85	6.03 ± 2.67	6.06 ± 2.24	0.15	**0.009 ***
Interquartile range (ML)	3.20 ± 1.73	3.43 ± 0.86	2.09 ± 1.50	3.14 ± 2.63	2.21 ± 1.20	1.91 ± 1.44	3.44 ± 1.21	3.77 ± 0.80	3.43 ± 1.55	3.68 ± 1.29	3.85 ± 1.74	4.32 ± 1.95	0.51	0.06
Autocorrelation (AP) ${\;}^{a}$	7.98 ± 5.84	7.06 ± 3.06	7.13 ± 6.72	7.83 ± 4.61	8.60 ± 6.29	11.15 ± 10.20	7.64 ± 5.87	10.99 ± 5.75	8.77 ± 4.73	12.54 ± 6.94	10.59 ± 4.46	8.90 ± 5.17	0.80	0.23
Autocorrelation (ML) ${\;}^{a}$	4.61 ± 3.48	6.42 ± 2.41	3.32 ± 2.89	3.55 ± 2.41	4.40 ± 3.00	3.01 ± 2.33	7.33 ± 4.05	4.81 ± 2.10	4.02 ± 2.69	4.18 ± 1.80	4.60 ± 2.10	4.99 ± 3.31	0.15	0.13
Signal distance (AP) ${\;}^{b}$	11.64 ± 1.62	13.54 ± 0.41	10.95 ± 3.234	12.70 ± 1.69	10.31 ± 3.23	12.64 ± 1.73	12.32 ± 0.18	13.44 ± 0.14	11.90 ± 1.17	13.457 ± 0.14	11.95 ± 1.26	13.42 ± 0.09	**0.001 ***	0.09
Signal distance (ML) ${\;}^{b}$	11.57 ± 1.60	13.37 ± 0.11	10.84 ± 3.22	12.68 ± 1.74	10.28 ± 3.28	12.59 ± 1.70	12.28 ± 0.17	13.41 ± 0.09	11.86 ± 1.23	13.39 ± 0.11	11.93 ± 1.26	13.41 ± 0.11	**0.001 ***	0.09

**Abbreviations:** WS (within subjects), BS (between subjects). * denotes an alpha level at *p* < 0.05, ${\;}^{a}$: $\times 10^{4}$; ${\;}^{b}$: $\times 10^{2}$.

**Table 7 sensors-24-04750-t007:** Results from Mixed ANOVA for statistically significant calculations.

Metric	Direction	Group	Week	Week × Group
Maximum	AP	$F_{1,14}$ = 1.483	$F_{5,70}$ = 0.705	$F_{5,70}$ = 1.505
		*p* = 0.243	*p* = 0.622	*p* = 0.200
		${n_{p}}^{2}$ = 0.096	${n_{p}}^{2}$ = 0.048	${n_{p}}^{2}$ = 0.097
	ML	$F_{1,14}$ = 2.951	**$F_{5,70}$ = 2.457**	$F_{5,70}$ = 2.141
		*p* = 0.108	* **p** * **= 0.041**	p = 0.070
		${n_{p}}^{2}$ = 0.174	**${n_{p}}^{2}$ = 0.149**	${n_{p}}^{2}$ = 0.133
Minimum	AP	**$F_{1,14}$ = 8.812**	$F_{5,70}$ = 1.886	$F_{5,70}$ = 0.107
		* **p** * **= 0.010**	*p* = 0.108	*p* = 0.991
		**${n_{p}}^{2}$ = 0.386**	${n_{p}}^{2}$ = 0.119	${n_{p}}^{2}$ = 0.008
	ML	$F_{1,14}$ = 0.451	$F_{3.592,50,282}$ = 0.902	$F_{3.592,50,282}$ = 0.456
		*p* = 0.513	*p* = 0.462	*p* = 0.748
		${n_{p}}^{2}$ = 0.031	${n_{p}}^{2}$ = 0.061	${n_{p}}^{2}$ = 0.032
Standard deviation	AP	**$F_{1,14}$ = 16.012**	$F_{5,70}$ = 1.431	$F_{5,70}$ = 1.429
		* **p** * **= 0.001**	*p* = 0.224	*p* = 0.225
		**${n_{p}}^{2}$ = 0.534**	${n_{p}}^{2}$ = 0.093	${n_{p}}^{2}$ = 0.093
	ML	$F_{1,14}$ = 3.748	$F_{5,70}$ = 1.078	$F_{5,70}$ = 1.853
		*p* = 0.073	*p* = 0.380	*p* = 0.114
		${n_{p}}^{2}$ = 0.211	${n_{p}}^{2}$ = 0.072	${n_{p}}^{2}$ = 0.117
Variance	AP	**$F_{1,14}$ = 14.310**	$F_{5,70}$ = 1.287	$F_{5,70}$ = 1.301
		* **p** * **= 0.002**	*p* = 0.279	*p* = 0.274
		**${n_{p}}^{2}$ = 0.505**	${n_{p}}^{2}$ = 0.084	${n_{p}}^{2}$ = 0.085
	ML	$F_{1,14}$ = 1.474	$F_{5,70}$ = 0.722	$F_{5,70}$ = 1.821
		*p* = 0.245	*p* = 0.609	*p* = 0.120
		${n_{p}}^{2}$ = 0.095	${n_{p}}^{2}$ = 0.049	${n_{p}}^{2}$ = 0.115
Root mean square	AP	$F_{1,14}$ = 4.047	$F_{5,70}$ = 0.365	$F_{5,70}$ = 0.331
		*p* = 0.064	*p* = 0.871	*p* = 0.893
		${n_{p}}^{2}$ = 0.224	${n_{p}}^{2}$ = 0.025	${n_{p}}^{2}$ = 0.023
	ML	**$F_{1,14}$ = 16.506**	$F_{5,70}$ = 1.828	$F_{5,70}$ = 1.676
		* **p** * **= 0.010**	*p* = 0.119	*p* = 0.152
		**${n_{p}}^{2}$ = 0.388**	${n_{p}}^{2}$ = 0.115	${n_{p}}^{2}$ = 0.107
Interquartile range	AP	**$F_{1,14}$ = 9.262**	$F_{5,70}$ = 1.676	$F_{5,70}$ = 0.959
		* **p** * **= 0.009**	*p* = 0.152	p = 0.449
		**${n_{p}}^{2}$ = 0.398**	${n_{p}}^{2}$ = 0.107	${n_{p}}^{2}$ = 0.064
	ML	$F_{1,14}$ = 4.049	$F_{5,70}$ = 0.868	$F_{5,70}$ = 1.788
		*p* = 0.064	*p* = 0.507	*p* = 0.127
		${n_{p}}^{2}$ = 0.224	${n_{p}}^{2}$ = 0.058	${n_{p}}^{2}$ = 0.113
Signal distance	AP	$F_{1,14}$ = 3.300	**$F_{2.023,28.323}$ = 8.586**	$F_{2.023,28.323}$ = 0.700
		*p* = 0.091	* **p** * **= 0.001**	*p* = 0.507
		${n_{p}}^{2}$ = 0.191	**${n_{p}}^{2}$ = 0.380**	${n_{p}}^{2}$ = 0.048
	ML	$F_{1,14}$ = 3.228	**$F_{2.011,28.156}$ = 8.708**	$F_{2.011,28.156}$ = 0.631
		*p* = 0.094	* **p** * **= 0.001**	*p* = 0.540
		${n_{p}}^{2}$ = 0.187	**${n_{p}}^{2}$ = 0.383**	${n_{p}}^{2}$ = 0.043
$H_{qs=-5}$	AP	**$F_{1,14}$ = 5.478**	$F_{1.668,23.359}$ = 0.707	$F_{1.668,23.359}$ = 0.521
		* **p** * **= 0.035**	*p* = 0.479	*p* = 0.569
		**${n_{p}}^{2}$ = 0.281**	${n_{p}}^{2}$ = 0.048	${n_{p}}^{2}$ = 0.036
	ML	$F_{1,14}$ = 4.495	$F_{1.648,23.067}$ = 0.667	$F_{1.648,23.067}$ = 0.946
		*p* = 0.052	*p* = 0.495	*p* = 0.387
		${n_{p}}^{2}$ = 0.243	${n_{p}}^{2}$ = 0.045	${n_{p}}^{2}$ = 0.063
$H_{qs=0}$	AP	**$F_{1,14}$ = 4.837**	$F_{2.167,30.344}$ = 1.651	$F_{2.167,30.344}$ = 0.375
		* **p** * **= 0.045**	*p* = 0.207	*p* = 0.707
		**${n_{p}}^{2}$ = 0.257**	${n_{p}}^{2}$ = 0.105	${n_{p}}^{2}$ = 0.026
	ML	$F_{1,14}$ = 3.736	$F_{2.196,30.741}$ = 1.179	$F_{2.196,30.741}$ = 1.152
		*p* = 0.074	*p* = 0.325	*p* = 0.333
		${n_{p}}^{2}$ = 0.211	${n_{p}}^{2}$ = 0.078	${n_{p}}^{2}$ = 0.076
$H_{qs=4.5}$	AP	**$F_{1,14}$ = 6.548**	$F_{2.340,32.762}$ = 1.651	$F_{2.340,32.762}$ = 0.218
		* **p** * **= 0.023**	*p* = 0.204	*p* = 0.838
		**${n_{p}}^{2}$ = 0.319**	${n_{p}}^{2}$ = 0.105	${n_{p}}^{2}$ = 0.015
	ML	$F_{1,14}$ = 4.478	$F_{2.842,39.794}$ = 1.262	$F_{2.842,39.794}$ = 0.764
		*p* = 0.053	*p* = 0.300	*p* = 0.514
		${n_{p}}^{2}$ = 0.242	${n_{p}}^{2}$ = 0.083	${n_{p}}^{2}$ = 0.052
Negative width	AP	**$F_{1,14}$ = 6.428**	$F_{3.447,48.257}$ = 0.826	$F_{3.447,48.257}$ = 1.174
		* **p** * **= 0.024**	*p* = 0.500	*p* = 0.332
		**${n_{p}}^{2}$ = 0.315**	${n_{p}}^{2}$ = 0.056	${n_{p}}^{2}$ = 0.077
	ML	**$F_{1,14}$ = 6.716**	$F_{3.023,42.325}$ = 0.937	$F_{3.023,42.325}$ = 0.937
		* **p** * **= 0.021**	*p* = 0.432	*p* = 0.432
		**${n_{p}}^{2}$ = 0.324**	${n_{p}}^{2}$ = 0.063	${n_{p}}^{2}$ = 0.063

**Abbreviations:** AP: anterior–posterior; ML: medial–lateral. The values highlighted in bold denote an alpha level at *p* < 0.05.

**Table 8 sensors-24-04750-t008:** Mean ± SD for calculated nonlinear measures with *p* values for within and between subjects comparison.

Metric	Day/Time												WS	BS
	**PG**						**CSPG**						* **p** *	* **p** *
	**Day 1**		**Day 3**		**Day 5**		**Day 1**		**Day 3**		**Day 5**			
	**AM**	**PM**	**AM**	**PM**	**AM**	**PM**	**AM**	**PM**	**AM**	**PM**	**AM**	**PM**		
$H_{qs=2}$ (AP)	1.54 ± 0.18	1.52 ± 0.11	1.61 ± 0.15	1.63 ± 0.16	1.52 ± 0.10	1.43 ± 0.15	1.58 ± 0.15	1.60 ± 0.11	1.50 ± 0.18	1.56 ± 0.12	1.56 ± 0.18	1.52 ± 0.17	0.34	0.82
$H_{qs=2}$ (ML)	1.50 ± 0.19	1.42 ± 0.16	1.37 ± 0.18	1.37 ± 0.15	1.37 ± 0.11	1.25 ± 0.18	1.40 ± 0.12	1.45 ± 0.11	1.50 ± 0.16	1.49 ± 0.16	1.40 ± 0.09	1.37 ± 0.12	0.07	0.16
$H_{qs=-5}$ (AP)	1.98 ± 0.26	2.06 ± 0.13	1.83 ± 0.54	1.95 ± 0.22	1.76 ± 0.58	1.92 ± 0.25	2.19 ± 0.13	2.15 ± 0.13	2.13 ± 0.35	2.15 ± 0.14	2.11 ± 0.31	2.49 ± 1.16	0.48	**0.04 ***
$H_{qs=-5}$ (ML)	1.95 ± 0.27	2.04 ± 0.08	1.80 ± 0.58	1.93 ± 0.30	1.73 ± 0.57	1.89 ± 0.23	1.94 ± 0.23	2.06 ± 0.13	2.20 ± 0.35	2.13 ± 0.18	2.09 ± 0.23	2.48 ± 1.2	0.50	0.05
$H_{qs=0}$ (AP)	1.63 ± 0.20	1.64 ± 0.08	1.50 ± 0.46	1.60 ± 0.21	1.41 ± 0.45	1.53 ± 0.19	1.71 ± 0.12	1.74 ± 0.08	1.63 ± 0.23	1.73 ± 0.09	1.66 ± 0.21	1.73 ± 1.00	0.21	**0.04 ***
$H_{qs=0}$ (ML)	1.56 ± 0.22	1.62 ± 0.07	1.43 ± 0.45	1.52 ± 0.27	1.35 ± 0.47	1.43 ± 0.15	1.57 ± 0.07	1.60 ± 0.08	1.60 ± 0.19	1.67 ± 0.11	1.59 ± 0.18	1.64 ± 0.07	0.33	0.07
$H_{qs=4.5}$ (AP)	1.36 ± 0.22	1.40 ± 0.06	1.29 ± 0.38	1.36 ± 0.19	1.21 ± 0.38	1.30 ± 0.13	1.50 ± 0.12	1.50 ± 0.08	1.37 ± 0.20	1.45 ± 0.07	1.38 ± 0.21	1.48 ± 0.08	0.20	**0.02 ***
$H_{qs=4.5}$ (ML)	1.28 ± 0.23	1.33 ± 0.07	1.13 ± 0.40	1.25 ± 0.27	1.11 ± 0.45	1.15 ± 0.11	1.33 ± 0.14	1.34 ± 0.10	1.35 ± 0.18	1.39 ± 0.10	1.30 ± 0.21	1.32 ± 0.08	0.30	0.05
Total width (AP)	0.94 ± 0.22	1.01 ± 0.21	0.80 ± 0.30	0.87 ± 0.19	0.79 ± 0.31	0.92 ± 0.20	1.01 ± 0.19	0.95 ± 0.18	1.08 ± 0.30	1.01 ± 0.16	1.05 ± 0.27	1.33 ± 1.15	0.57	0.10
Total width (ML)	0.98 ± 0.17	1.03 ± 0.08	0.97 ± 0.31	1.00 ± 0.26	0.89 ± 0.30	1.08 ± 0.34	0.95 ± 0.22	1.05 ± 0.12	1.19 ± 0.34	1.06 ± 0.13	1.11 ± 0.23	1.49 ± 1.26	0.39	0.17
Positive width (AP)	0.45 ± 0.11	0.43 ± 0.16	0.36 ± 0.17	0.40 ± 0.14	0.33 ± 0.16	0.39 ± 0.16	0.38 ± 0.12	0.40 ± 0.15	0.48 ± 0.19	0.46 ± 0.11	0.46 ± 0.17	0.70 ± 0.98	0.59	0.27
Positive width (ML)	0.46 ± 0.10	0.46 ± 0.05	0.49 ± 0.16	0.45 ± 0.23	0.39 ± 0.19	0.47 ± 0.25	0.37 ± 0.18	0.44 ± 0.11	0.48 ± 0.21	0.46 ± 0.08	0.47 ± 0.14	0.80 ± 1.04	0.46	0.52
Negative width (AP)	0.49 ± 0.13	0.58 ± 0.09	0.45 ± 0.13	0.47 ± 0.07	0.46 ± 0.17	0.53 ± 0.12	0.63 ± 0.12	0.55 ± 0.10	0.60 ± 0.15	0.55 ± 0.10	0.60 ± 0.13	0.63 ± 0.18	0.50	**0.02 ***
Negative width (ML)	0.52 ± 0.15	0.57 ± 0.08	0.49 ± 0.20	0.55 ± 0.09	0.50 ± 0.15	0.61 ± 0.11	0.58 ± 0.10	0.61 ± 0.13	0.70 ± 0.16	0.60 ± 0.10	0.64 ± 0.14	0.70 ± 0.23	0.43	**0.02 ***
Maximum spectrum (AP)	1.66 ± 0.20	1.68 ± 0.09	1.53 ± 0.47	1.64 ± 0.21	1.45 ± 0.46	1.57 ± 0.20	1.76 ± 0.12	1.78 ± 0.08	1.72 ± 0.27	1.78 ± 0.09	1.71 ± 0.22	2.06 ± 0.98	0.46	0.06
Maximum spectrum (ML)	1.60 ± 0.22	1.66 ± 0.07	1.47 ± 0.46	1.56 ± 0.27	1.39 ± 0.47	1.48 ± 0.16	1.56 ± 0.20	1.65 ± 0.09	1.70 ± 0.24	1.71 ± 0.12	1.64 ± 0.18	1.98 ± 0.99	0.53	0.09
Sample Entropy (AP)	0.42 ± 0.07	0.46 ± 0.10	0.41 ± 0.13	0.45 ± 0.07	0.39 ± 0.14	0.48 ± 0.13	0.46 ± 0.06	0.45 ± 0.05	0.41 ± 0.06	0.44 ± 0.03	0.40 ± 0.08	0.43 ± 0.06	0.19	0.87
Sample Entropy (ML)	0.43 ± 0.07	0.41 ± 0.06	0.39 ± 0.11	0.42 ± 0.05	0.37 ± 0.11	0.49 ± 0.18	0.48 ± 0.08	0.46 ± 0.05	0.42 ± 0.06	0.46 ± 0.04	0.44 ± 0.08	0.48 ± 0.04	0.08	0.09

**Abbreviations:** WS (Within Subjects), BS (Between Subjects). * denotes an alpha level at *p* < 0.05.

## Data Availability

The data presented in this study are available on request from the corresponding author.

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
