# Peer review of "Exploring the Real-Time Variability and Complexity of Sitting Patterns in Office Workers with Non-Specific Chronic Spinal Pain and Pain-Free Individuals"

_sensors, 2024, doi:10.3390/s24144750_

Round 1
Reviewer 1 Report
Comments and Suggestions for Authors
This manuscript requires substantial revisions to be deemed suitable for publication. I have several major and minor comments for the authors to address.
Specific comments:
1. The content in the Introduction is too redundant. The first three paragraphs can be merged into one, summarizing the characteristics of work and posture in sedentary populations, and highlighting the high incidence of lower back pain as revealed by epidemiological studies.
2. Also, the Introduction section does not give a proper overview of the field, it would be good to align the citations more closely with the scope and objectives of the paper.
3. Does CSP include LBP? What is the difference between them? The introduction section's specific research on variability only involves LBP and does not mention CSP.
4. Method: Line 189, There is no clear definition of “mag” in the text; it only appears in the formula. It should be specifically explained in the text.
5. Has the estimation method for COP been validated?
Author Response
Dear Dr. Reviewer,
We would like to thank you for the observations. We have considered all the suggestions, listed them, and highlighted them in yellow.
Sincerely
REVIEWS
Comments 1: “This manuscript requires substantial revisions to be deemed suitable for publication. I have several major and minor comments for the authors to address.
Specific comments:
- The content in the Introduction is too redundant. The first three paragraphs can be merged into one, summarizing the characteristics of work and posture in sedentary populations, and highlighting the high incidence of lower back pain as revealed by epidemiological studies.”
Response 1: We would like to thank you for the valuable feedback. We have revised the Introduction section accordingly. The first three paragraphs have been merged into a single, concise paragraph. This new paragraph effectively summarizes the characteristics of work and posture in sedentary populations while emphasizing the high incidence of lower back pain, as highlighted by epidemiological studies. Additionally, we have synthesized the content to improve clarity and coherence (page number: 1 paragraph: 1 line: 18).
Original version
Chronic spinal pain (CSP) is a significant global health problem that affects millions of people worldwide\cite{bevan2015economic}. It is a well-established leading cause of disability, with a profound impact on the physical and mental well-being of individuals \cite{herman2019healthcare, friedly2010epidemiology, wuglobal}. Low back and neck pain are the most prevalent types of chronic pain, causing a considerable burden on society and healthcare systems \cite{herman2019healthcare, friedly2010epidemiology, bevan2015economic, sung2024postural}. Recent research, considering the Global Burden of Diseases, Injuries, and Risk Factors Study (GBD) data from 1990-2020, estimates that in 2020, 619 million (554 to 694) people reported having low back pain, and that in 2050 this number will probably increase about 36.4\% (29.9 to 43.2\%), being approximately 843 million people worldwide (759 to 933) \cite{ferreira2023global}. Despite the prevalence of CSP, effective treatment options remain limited, and management can be challenging due to the complex nature of this condition \cite{falla2017individualized, briggs2009prevalence, heredia2022pain, bevan2015economic}. Therefore, identifying risk factors appears to be of high importance for designing and implementing suitable prevention programs. Interestingly, among the identified modifiable risk factors in GBD, occupational conditions, such as ergonomic factors, contribute significantly for the years lived with disability \cite{ferreira2023global}. Office-based job roles that require prolonged sitting, such as public administration tax authorities, increase the risk of developing CSP\cite{issever2008depression}. Tax authorities are responsible for various operational tasks, such as tax collection and regulation, imposition of tariffs, and combating tax evasion and fraud, which require extended periods of sitting. This can lead to heightened discomfort \cite{issever2008depression, baker2018short, sondergaard2010variability}, resulting in changes in motor variability \cite{heredia2022pain, sondergaard2010variability, bibbo2019sitting}. In addition, tax authority workers experience high levels of stress, which can affect their work performance, leading to reduced job satisfaction, health-related lost productivity, increased absenteeism, and presenteeism \cite{pohling2016work}. These biopsychosocial risks can contribute to the onset or aggravation of ongoing CSP. Therefore, there is a rising concern about investigating occupational diseases in these workers.
Revised version
Chronic spinal pain (CSP), which includes low back pain (LBP), neck pain, and thoracic spinal pain, is a widespread global health issue \cite{briggs2009prevalence}, impacting millions of individuals \cite{bevan2015economic}. CSP is a leading cause of disability, with LBP and neck pain being the most prevalent forms \cite{cohen2021chronic, herman2019healthcare, friedly2010epidemiology, wuglobal}. Recent research indicates that an estimated 619 million people experienced LBP in 2020, with projections suggesting an increase to 843 million by 2050 \cite{ferreira2023global}. Job roles involving prolonged sitting posture, such as those in tax authorities, have been found to increase the risk of developing CSP due to ergonomic factors \cite{issever2008depression, ferreira2023global, heredia2022pain}. Tax authorities are engaged in tasks that necessitate extended periods of sitting posture, leading to heightened discomfort and resulting in changes in motor variability \cite{issever2008depression, baker2018short, sondergaard2010variability}. Additionally, tax authority workers experience high levels of stress, which can affect their work performance, leading to reduced job satisfaction, health-related lost productivity, and increased absenteeism. These biopsychosocial risks can contribute to the onset or aggravation of ongoing CSP\cite{pohling2016work}. Hence, there is a growing concern regarding investigating occupational diseases in these workers.
Comments 2: “2. Also, the Introduction section does not give a proper overview of the field, it would be good to align the citations more closely with the scope and objectives of the paper.”
Response 2: We would like to thank you for the valuable feedback. We have revised the introduction section to better align the evidence with the research objectives (page number: 2, paragraph: 2, line: 48):
Original version
Previous research has revealed that prolonged sitting can contribute to an increase in the variability of displacement of the center of pressure (COP) and lumbar curvature in the anterior-posterior direction. This means that over time, the standard deviation or variability of COP displacement and lumbar curvature tends to increase, while the complexity represented by sample entropy decreases, resulting in perceived discomfort \cite{sondergaard2010variability}. Furthermore, investigations into trunk postural control in individuals with LBP using an unstable surface have revealed compromised control in comparison to those without LBP. A recent systematic review and meta-analysis has indicated that LBP patients have inferior trunk postural control, as evidenced by various metrics, including higher root mean square displacement. The study suggests that impaired postural control in LBP patients may result from delayed or inaccurate corrective responses, leading to increased seat movements and potential trunk stiffening \cite{alshehri2024trunk}.
Revised version
Previous research has revealed that prolonged sitting increases the variability of displacement of the center of pressure (COP) and lumbar curvature in the anterior-posterior direction, while decreasing complexity represented by sample entropy, leading to discomfort \cite{sondergaard2010variability}. Studies on trunk postural control using an unstable surface indicate that individuals with LBP have inferior control compared to those without LBP, as shown by higher root mean square displacement \cite{alshehri2024trunk}. This impaired postural control may result from delayed or inaccurate corrective responses, increasing seat movements and potential trunk stiffening. A systematic review found that individuals with chronic LBP have delayed muscle onsets in response to perturbations, with inconclusive evidence on COP or kinematic differences compared to healthy controls \cite{knox2018anticipatory}. Additionally, patients with non-specific neck pain or whiplash-associated disorder exhibit greater postural instability, linked to proprioceptive impairment rather than pain duration \cite{ruhe2011altered}. Older adults with neck pain show diminished functional performance, as indicated by wavelet analysis (but not for entropy) measures, compared to healthy controls \cite{quek2014new}. It is interesting to note that both experimental and chronic neck-shoulder pain can increase variability in task timing, kinematics, and muscle activation during repetitive arm movement \cite{madeleine2008changes}. This suggests that variability plays an important role in motor control and may contribute to the transition from acute to chronic pain. During the acute stage, the central nervous system seeks the least painful biomechanical solution, while in the chronic stage, the chosen solutions are characterized by reduced flexibility of the motor system \cite{madeleine2008changes}.
Comments 3.1: “3. Does CSP include LBP? What is the difference between them?”
Response 3.1: We appreciate the reviewer’s observation regarding the types of CSP. We added the sentence to clarify, Chronic Spinal Pain encompasses various forms of spinal pain, including low back pain (LBP), neck pain, and thoracic pain (page number: 1, paragraph: 1, line: 18):
"Chronic spinal pain (CSP), which includes low back pain (LBP), neck pain, and thoracic spinal pain, is a widespread global health issue \cite{briggs2009prevalence}, impacting millions of individuals \cite{bevan2015economic}."
Comments 3.2: “The introduction section's specific research on variability only involves LBP and does not mention CSP.”
Response 3.2: We added a new sentence to illustrate other forms of CSP besides LPB (page number: 2, paragraph: 3, line: 57):
Additionally, patients with non-specific neck pain or whiplash-associated disorder exhibit greater postural instability, linked to proprioceptive impairment rather than pain duration \cite{ruhe2011altered}. Older adults with neck pain show diminished functional performance, as indicated by wavelet analysis (but not for entropy) measures, compared to healthy controls \cite{quek2014new}. It is interesting to note that both experimental and chronic neck-shoulder pain can increase variability in task timing, kinematics, and muscle activation during repetitive arm movement \cite{madeleine2008changes}. This suggests that variability plays an important role in motor control and may contribute to the transition from acute to chronic pain. During the acute stage, the central nervous system seeks the least painful biomechanical solution, while in the chronic stage, the chosen solutions are characterized by reduced flexibility of the motor system \cite{madeleine2008changes}.
Comments 4: “4. Method: Line 189, There is no clear definition of “mag” in the text; it only appears in the formula. It should be specifically explained in the text.
Response 4: We appreciate the reviewer’s observation regarding the clarity of the term "mag" in the text. To address this, we have revised the relevant section to include a detailed explanation of "mag” (page number: 5, paragraph: ‘2. Materials and Methods/ 2.4.2. Accelerometer and Rotation Vector Pre-Processing’, line: 185):
Original version
An algorithm was used to detect and remove periods when workers were not at their desks or transitioning between postures that were outside the analysis range \cite{ohlendorf2020standard}. This algorithm was fed by ACC data and considered a threshold to identify postural changes and walking intervals. The threshold was based on the acceleration magnitude from the three axes (X, Y, and Z). Figure \ref{fig:signal.png} displays the signal with green highlights indicating the removed parts.
Revised version
An algorithm was used to detect and remove periods when workers were not at their desks or transitioning between postures that were outside the analysis range \cite{ohlendorf2020standard}. This algorithm was fed by ACC data and considered a threshold to identify postural changes and walking intervals. The threshold was determined based on the magnitude of acceleration (mag) from the X, Y, and Z axes. The magnitude (mag) represents the combined acceleration value along the x, y, and z axes. In Figure \ref{fig:signal.png}, the graph shows the signal with green highlights indicating the segments that were removed.
Comments 5: “5. Has the estimation method for COP been validated”
Response 5: We would like to thank for the constructive feedback. The method we employed aligns with the established principles as described in the literature. We added a sentence in the Discussion section (page number: 13, paragraph: ‘Discussion/4.4. Implications and Future Research’, line: 473):
“Finally, future studies should consider comparing the COP estimation method. Although based on established physical principles and developed internally by our research team, future investigations should proceed with validation analysis through comparative studies or external benchmarks.”

Reviewer 2 Report
Comments and Suggestions for Authors
The reviewer understood that the present article mentioned potential causes of Chronic Spinal Pain of office workers with several wearable sensors. In order to improve of the quality of the article, I have some comments as follows:
(1) Please describe the hypothesis clearly in the main text
I found hypothesis description in the abstract, not in main text. Please mention the hypothesis in the main text clearly.
(2) Number of participants with CSP is different in abstract (11) and the main text (10)
In the abstract, number of participants was described as 11, whereas that was 10 in the main text. Please check it and modified it.
(3) Use period [.] instead of comma[,]
Line 193: C7 vertebrae (ax = 1,02 m/s2; ay = 1,15 m/s2; az = 1,41 m/s2)
(4) Line 195: "2 m/s2", 2 should be the superior letter
(5) Discussion section is too long, please set sub-clause
In this article, lots of indices were defined and calculated. In discussion section, authors discussed potential causes of CPS based on statistical results of indices. It is difficult to follow the holistic structure of results. I recommend to set sub-clause in the discussion section to clarify the holistic structure of results towards the clarification of hypothesis.
Author Response
Dear Dr. Reviewer,
Dear Dr. Reviewer,
We would like to thank you for the observations. We have considered all the suggestions, listed them, and highlighted them in yellow.
Sincerely
REVIEWER 2
Comments 1: “The reviewer understood that the present article mentioned potential causes of Chronic Spinal Pain of office workers with several wearable sensors. In order to improve of the quality of the article, I have some comments as follows: (1) Please describe the hypothesis clearly in the main text I found hypothesis description in the abstract, not in main text. Please mention the hypothesis in the main text clearly.”
Response 1: We appreciate the reviewer’s observation regarding the hypothesis. We have clarified it in the main text, explicitly stating it in the final paragraph of the Introduction (page number: 3, paragraph: 5 of the Introduction section, line: 86):
"The study had two main objectives: (a) to assess individual trunk motor variability of office workers in real-time and (b) to test the hypothesis that individuals with and without pain will exhibit different responses to office work tasks."
Comments 2: “(2) Number of participants with CSP is different in abstract (11) and the main text (10) In the abstract, number of participants was described as 11, whereas that was 10 in the main text. Please check it and modified it.”
Response 2: We appreciate the reviewer’s attention to this discrepancy. We acknowledge that the number of participants with CSP was incorrectly reported. The correct number is 10, and we have amended this in the abstract to ensure consistency with the main text. The revised sentence reads (page number: 1, paragraph: 1 of the abstract, line: 8):
“Participated in this study 6 office workers without pain and 10 with CSP.”
Comments 3: “(3) Use period [.] instead of comma[,]
Line 193: C7 vertebrae (ax = 1,02 m/s2; ay = 1,15 m/s2; az = 1,41 m/s2)”
Response 3: We appreciate the reviewer's attention to detail and assistance in improving our manuscript. We have made the suggested changes to the characters. The corrected line now reads (page number: 5, paragraph: ‘2. Materials and Methods/ 2.4.2. Accelerometer and Rotation Vector Pre-Processing’, line: 191):
C7 vertebrae (ax = 1.02 m/s²; ay = 1.15 m/s²; az = 1.41 m/s²).
Comments 4: “(4) Line 195: "2 m/s2", 2 should be the superior letter”
Response 4: We appreciate the reviewer's valuable feedback and have changed the characters as suggested. The corrected line now reads: "2 m/s²," with the "2" as a superscript (page number: 5, paragraph: ‘2. Materials and Methods/ 2.4.2. Accelerometer and Rotation Vector Pre-Processing’, line: 200).
Comments 5: “(5) Discussion section is too long, please set sub-clause”
“In this article, lots of indices were defined and calculated. In discussion section, authors discussed potential causes of CPS based on statistical results of indices. It is difficult to follow the holistic structure of results. I recommend to set sub-clause in the discussion section to clarify the holistic structure of results towards the clarification of hypothesis.”
Response 5: We appreciate the valuable feedback and have incorporated the suggestions to improve the clarity and readability of the discussion section. The discussion is now organized into the following subsections (page number: 11, paragraph: Discussion section):
- 4.1. Statistical Domain Metrics (line: 366)
- 4.2. Temporal Domain Metrics (line: 380)
- 4.3. Signal Structure and Complexity (line: 389)
- 4.4. Implications and Future Research (line: 447)
